

# Sensor-Independent LAI/FPAR CDR: Reconstructing a Global Sensor-Independent Climate Data Record of MODIS and VIIRS LAI/FPAR from 2000 to 2022

Jiabin Pu[1], Kai Yan[2], Samapriya Roy[3], Zaichun Zhu[4], Miina Rautiainen[5], Yuri Knyazikhin[1], Ranga B.
Myneni[1]

[1]Department of Earth and Environment, Boston University, Boston, MA 02215, USA
[2]Faculty of Geographical Science, Beijing Normal University, Beijing, 100875, China
[3]University of Arizona, Tucson, AZ 85721, USA
[4]School of Urban Planning and Design, Shenzhen Graduate School, Peking University, Shenzhen 518055, China
[5]Aalto University, School of Engineering, P.O. Box 14100, FI-00076 Aalto, Finland

*Correspondence to*: Kai Yan (kaiyan@bnu.edu.cn)

**Abstract.** Leaf area index (LAI) and fraction of photosynthetically active radiation (FPAR) are critical biophysical parameters for the characterization of terrestrial ecosystems. Long-term global LAI/FPAR products, such as MODIS&VIIRS, provide the fundamental dataset for accessing vegetation dynamics and studying climate change. However, existing global LAI/FPAR
products suffer from several limitations, including spatial-temporal inconsistencies and accuracy issues. Considering these limitations, this study develops a Sensor-Independent (SI) LAI/FPAR climate data record (CDR) based on Terra-MODIS/Aqua-MODIS/VIIRS LAI/FPAR standard products. The SI LAI/FPAR CDR covers the period from 2000 to 2022, at spatial resolutions of 500m/5km/0.05 degrees, 8-day/bimonthly temporal frequencies and available in sinusoidal and WGS1984 projections. The methodology includes (*i*) comprehensive analyses of sensor-specific quality assessment variables
to select high quality retrievals, (*ii*) application of the spatial-temporal tensor (ST-Tensor) completion model to extrapolate LAI and FPAR beyond areas with high quality retrievals, (*iii*) generation of SI LAI/FPAR CDR in various projections, spatial and temporal resolutions, and (*iv*) evaluation of the CDR by direct comparisons to ground data and indirectly through reproducing results of LAI/FPAR trends documented in literature. This paper provides a comprehensive analysis of each step involved in the generation of the SI LAI/FPAR CDR, as well as evaluation of the ST-Tensor completion model. Comparisons
of SI LAI (FPAR) with ground truth data suggest a RMSE of 0.84 LAI (0.15 FPAR) units with $R^2$ of 0.72 (0.79), which are improvements of the standard Terra/Aqua/VIIRS LAI (FPAR) products by 0.02~0.19 LAI (0.01~0.02 FPAR) units with the $R^2$ decreased by 0.02~0.16 (0.05~0.09). The SI LAI/FPAR CDR is characterized by a low time series stability (TSS) value, suggesting a more stable and less noisy data set than their sensor-dependent counterparts. Furthermore, the mean absolute error (MAE) of the CDR is also lower, suggesting that SI LAI/FPAR CDR is comparable in accuracy with high-quality retrievals.
LAI/FPAR trend analyses based on the SI LAI/FPAR CDR agrees with previous studies, which indirectly provides enhanced capabilities to utilize this CDR for studying vegetation dynamics and climate change. Overall, the integration of multiple satellite data sources and the use of advanced gap-filling modelling techniques improve the accuracy of the SI LAI/FPAR



CDR, ensuring the reliability of long-term vegetation studies, global carbon cycle modelling and land policy development for informed decision-making and sustainable environmental management.

**Keywords:** MODIS, VIIRS, LAI, FPAR, Sensor-Independent, CDR.

# 1 Introduction

The leaf area index (LAI) is a fundamental parameter for quantifying the structural and functional characteristics of terrestrial vegetation canopies, defined as half of the total green foliage per unit of horizontal ground area (Chen and Black, 1992;Chen, 1996). LAI plays an essential role in models of ecological processes, global primary productivity, climate

dynamics, water cycle and the carbon cycle analysis (Sellers et al., 1997;Boussetta et al., 2013;Piao et al., 2015;Fang et al., 2019;Chen et al., 2022). The fraction of incident photosynthetically active radiation (400-700 nm) absorbed by vegetation (FPAR) is an important biophysical parameter used to quantify the energy absorption capacity of the vegetation canopy (Knyazikhin et al., 1998a;Myneni et al., 2002). LAI and FPAR are key climate variables and biodiversity metrics identified by the United Nations Global Climate Observing System (GCOS)(Skidmore et al., 2015).

In recent decades, there has been a remarkable increase in the use of global long-term satellite-derived LAI/FPAR datasets from various sensors, e.g., the Advanced Very High-Resolution Radiometer (AVHRR), the Moderate Resolution Imaging Spectroradiometer (MODIS), and the Visible Infrared Imager Radiometer Suite (VIIRS). Among these, the LAI/FPAR products from MODIS on the Terra platform have been widely used since 2000 and represent a milestone in operational generation of vegetation parameters from satellite observations (Knyazikhin, 1999;Myneni and Park, 2015;Yan et al.,

2016;Yan et al., 2021c). The LAI/FPAR are also available from MODIS on the Aqua platform and VIIRS on the Suomi National Polar-Orbiting Partnership (S-NPP) and the Joint Polar Satellite System (JPSS) satellites since 2002 (2012), ensuring the extension of the Terra MODIS long-term data record (Justice et al., 2013). The MODIS&VIIRS LAI/FPAR datasets have contributed significantly to many studies, such as terrestrial carbon sinks, understanding seasonal and interannual variations in equatorial forests, analyses of spatial patterns of drought, and climate and energy flux dynamics (Tang et al., 2013;Mariano

et al., 2018;Chen et al., 2019;Chen et al., 2022;Sun et al., 2022).

Two weaknesses of the MODIS/VIIRS LAI/FPAR products have been identified (i.e., temporal stability and absolute accuracy), which limit their application in vegetation dynamic studies (Fang et al., 2012a;Fang et al., 2019;Yan et al., 2021a). These problems mainly arise due to uncertainties in input information to the operational retrieval algorithm such as surface reflectance and land cover type (Knyazikhin, 1999;Fang et al., 2019;Tian et al., 2000). Several post-processing techniques

have been proposed to remove uncertainties in the MODIS&VIIRS LAI/FPAR standard products. These include 1) identifying areas with a high fraction of water area in the satellite pixel and removing their impact on the retrieval using a mixed pixel correction method (Xu et al., 2020;Dong et al., 2023); 2) integration of prior knowledge of reflectance variations into the generation of the image composite (Pu et al., 2023); 3) accounting for the canopy hot spot effect in the retrieval technique (Yan et al., 2021b). These methods would increase product spatial coverage and developing various gap filling techniques to

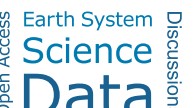

extrapolate retrievals beyond areas with valid satellite observations such as 1) cubic splines (Mitášová and Hofierka, 1993); 2) spatial linear, bilinear and kriging interpolations (Xu et al., 2015;Smith, 1981;Oliver and Webster, 1990), 3) various temporal extrapolation techniques (Holben, 1986;Lange et al., 2017;Roerink et al., 2000;Zhu et al., 2011;Das and Ghosh, 2017;Chu et al., 2021;Wang et al., 2023). However, most of the approaches are characterized by high computational costs and/or lack of information necessary for their implementation at the global scale. Consequently, most research has been limited to specific

regions (e.g., China and North America), leaving a significant gap in developing Climate Data Record (CDR), which by definition is a consistently-processed time series of uncertainty-quantified data, located in time and space, of sufficient length and quality to be useful for climate time-scale uses (Merchant et al., 2017).

The objective of this paper is to develop a long-term LAI and FPAR CDR using standard LAI/FPAR products from MODIS and VIIRS sensors. Our approach includes (*i*) comprehensive analyses of sensor-specific quality assessment (QA)

variables to select high-quality retrievals, (*ii*) application of the spatial-temporal tensor (ST-Tensor) completion model to extrapolate LAI and FPAR beyond areas with high quality retrievals, (*iii*) generation of SI LAI/FPAR CDRs in various projections, spatial and temporal resolutions, and (*iv*) evaluation of the CDR by direct comparisons to ground data and indirectly through reproducing results of LAI/FPAR trend analyses to revisit the Greening Earth.

The paper is organized as follows. Section 2 introduces the data used in this study and the study area. Section 3 details

the steps involved in generating the SI LAI/FPAR CDR, including analyses of sensor-specific QA, generation of high-quality Filtered SI LAI/FPAR timeseries, and applying ST-Tensor completion model. The results of the validation and evaluation are presented in Section 4. Section 5 discusses the underlying factors that contributed to the improvement of the SI LAI/FPAR CDR and the associated issues and challenges. The final section concludes the paper by summarizing the key findings and highlighting the significance of the research.

## 2 Datasets and Study Area

### 2.1 LAI/FPAR Products: MOD15A2H, MYD15A2H, and VNP15A2H

The MODIS&VIIRS LAI/FPAR inputs surface bidirectional reflectance factors (BRFs) in the red and near-infrared (NIR) spectral bands, their uncertainties, sun-sensor geometry, and biome classification map and retrieves the LAI and FPAR for each satellite pixel. The retrieval technique consists of a main algorithm which is based on the radiative transfer equation (RTE)

and a backup algorithm, which uses empirical relationships between normalized difference vegetation index (NDVI) and canopy LAI/FPAR (Myneni et al., 2002;Yan et al., 2018;Knyazikhin, 1999;Knyazikhin et al., 1998b). The main algorithm compares the observed spectral BRF with those evaluated from the RTE-based entries stored in a look-up-table (LUT) for a suite of canopy structures and soil patterns that represent an expected range of typical conditions for a given biome type. All canopy/soil patterns for which modelled and observed BRFs differ within a specified uncertainty level are considered as

acceptable solutions. The mean values of LAI and FPAR and their dispersions are reported as retrievals and their uncertainties. When this method fails to localize a solution, the backup method is utilized (Myneni et al., 2002;Yan et al., 2018).



Performance analyses of the MODIS LAI/FPAR algorithm indicate that the best quality, high-precision retrievals are obtained from the main algorithm (Yan et al., 2021a). In the case of dense canopies, the BRFs saturate and become weakly sensitive to changes in canopy properties. The reliability of parameters retrieved under the condition of saturation is lower than

that generated by the main algorithm using unsaturated BRF. Such retrievals are flagged. The algorithm path is the key quality assessment (QA) flag that provides information about the overall quality of the LAI/FPAR. It includes four values (from highest to lowest quality): the main algorithm without saturation, the main algorithm with saturation, the backup algorithm due to sun-sensor geometry, and the backup algorithm due to other reasons. The QA variables, FparLai_QC and FparExtra_QC, provide information about the cloud state, aerosol load, and the presence of snow, which are inherited from the upstream BRF

product (Knyazikhin, 1999;Myneni and Park, 2015;Yan et al., 2016;Park et al., 2017).

The daily retrievals are composited over an 8-day period by selecting the LAI and FPAR pair corresponding to the maximum FPAR value generated by the main algorithm (Knyazikhin and Myneni, 2018). The backup algorithm retrievals are selected only when no main algorithm retrievals are available during the 8-day compositing period. The 8-day composited LAI/FPAR product is distributed to the public from the NASA EOSDIS Land Processes Distributed Active Archive Center

(Myneni et al., 2015;Myneni and Knyazikhin, 2018).

In this study, we used the Collection 6 (C6) MOD15A2H, C6 MYD15A2H, and the Collection (C1) VNP15A2H LAI/FPAR products. The products are available at 500m sinusoidal grid and are updated every 8 days, resulting in approximately 46 composites per year. MOD15A2H data have been available since 18 February 2000, MYD15A2H since 14 July 2002 and VNP15A2H since 17 January 2012. These datasets are distributed in standard hierarchical data format (HDF)

files. Each HDF file contains six scientific data sets (SDS): FPAR, LAI, FparLai_QC, FparExtra_QC, FparStdDev, and LaiStdDev. The LAI and FPAR layers contain the LAI/FPAR retrievals, while the FparLai_QC and FparExtra_QC layers provide information about the algorithm paths and atmospheric conditions. This quality information underlies the first step in generating SI LAI/FPAR CDR, as detailed in Section 3.1.

## 2.2 Land Cover Map: MCD12Q1

The MODIS land cover product (MCD12Q1) provides a global map of land cover types at a spatial resolution of 500 meters and an annual time step (Sulla-Menashe and Friedl, 2018). In this study, we used the LAI legacy classification scheme (LC_Type_3), which categorizes global vegetation into eight biomes: grasses and cereal crops (Biome 1), shrubs (Biome 2), broadleaf crops (Biome 3), savannas (Biome 4), Evergreen Broadleaf Forests (EBF, Biome 5), Deciduous Broadleaf Forests (DBF, Biome 6), Evergreen Needleleaf Forests (ENF, Biome 7), and Deciduous Needleleaf Forests (DNF, Biome 8) (Sulla-

Menashe and Friedl, 2018). The biome classification map is an important auxiliary dataset for MODIS&VIIRS LAI/FPAR operational algorithms. It reduces the number of unknowns of the inverse problem through the use of simplifying assumptions (e.g., leaf normal orientation) and standard constants (e.g., leaf albedo, patterns of ground reflectance) that are assumed to vary with the biome (Knyazikhin, 1999). A global distribution of 8 biomes is shown in Fig. 1.



### 2.3 Ground LAI/FPAR Reference

The growing utilization of Earth observation (EO) products has highlighted the importance of addressing product uncertainty through validation based on ground measurements (Baret et al., 2006;Yang et al., 2006;Fang et al., 2012b). In our study, we validated our SI LAI/FPAR CDR by comparing their values with LAI and FPAR ground reference data from version 3 Copernicus Ground-Based Observations for Validation (GBOV) and version 2 DIRECT database (Brown et al., 2020;Brown et al., 2021). The combined utilization of both sets of measurment provides comprehensive coverage globally, encompassing

a diverse array of representative biome types. The only notable exceptions are the eastern part of China and Eastern Europe, where the absence of measurement sites.

        GBOV, part of the Copernicus Global Land Service, aims to facilitate the use of ground-based observations to validate EO products and ensure their quality and consistency. GBOV collects multi-year ground-based observations from global networks and upgrades existing sites or establishes new ones to bridge thematic or geographic gaps (Brown et al., 2020;Bai et

al., 2022). To ensure data quality, the GBOV reference database has undergone rigorous quality control procedures and includes various measurements such as top of canopy reflectance, surface albedo, LAI, FPAR, proportion of ground cover, soil moisture at 5 cm depth, and surface temperature. GBOV data are available through the open GBOV portal (https://gbov.acri.fr). We used the GBOV LAI/FPAR maps since 2014 as a reference dataset. We selected 29 GBOV sites of 3 km x 3 km each (see Fig. 1). The LAI/FPAR reference maps were averaged over the 3 km x 3 km area for product assessment.

Statistics based on downloaded data, the GBOV validation data set used in this study consisted of 9805 LAI and 10548 FPAR measurements.

        The DIRECT LAI, FPAR, and vegetation cover are available as spatially-average values over 3km x 3km reference maps (Brown et al., 2021). Following the CEOS WGCV LPV good practice, the ground data are enhanced with high spatial resolution imagery to account for spatial heterogeneity. This dataset includes 176 global sites representing 7 major biome types,

covering the period from 2000 to 2021. Forest sites without understorey are filtered out of the DIRECT database (https://calvalportal.ceos.org/lpv-direct-v2.1). The dataset used in this study contains 446 LAI and 109 FPAR measurements.

### 2.4 Study Area

        Fig. 1 shows distribution of 8 biomes and selected GBOV and DIRECT validation sites. We selected a study area of global, and selected the typical area of 86°W to 30°W and from 20°S to 10°N (zoom-in case in Fig. 1), containing Amazonian

forests to highlight the importance of various steps in developing the SI LAI/FPAR CDR. Obtaining high-quality observations over this area is difficult due to a large amount of cloud-contaminated data. It is therefore particularly valuable to assess the quality of our LAI/FPAR CDR in this region.

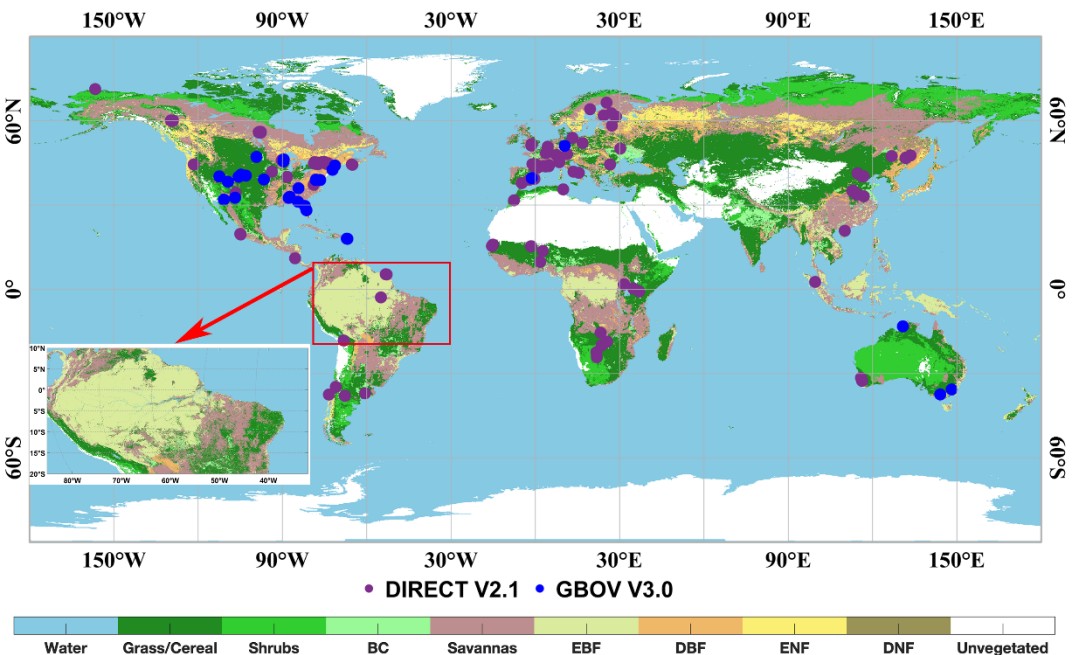

**Figure 1.** Distribution of the selected GBOV and DIRECT sites. Amazonian forests used as a study area to assess various steps in developing
LAI/FPAR CDR is shown as a red square. The background color indicates the biome types from the MCD12Q1 classification schemes of
the year 2017 (grasses and cereal crops (Biome 1), shrubs (Biome 2), broadleaf crops (Biome 3), savannas (Biome 4), Evergreen Broadleaf
Forests (EBF, Biome 5), Deciduous Broadleaf Forests (DBF, Biome 6), Evergreen Needleleaf Forests (ENF, Biome 7), and Deciduous
Needleleaf Forests (DNF, Biome 8)). The blue and purple dots represent the GBOV3.0 and DIRECT2.1 sites.

## 2.5 Metrics for Evaluating

In this study, we assessed the variability of SI LAI/FPAR CDR using two metrics: time series stability (TSS) (Weiss et
al., 2007;Zou et al., 2022) and mean absolute error (MAE) (Zhou et al., 2015). Both metrics provide insight into the
uncertainties associated with a dataset. TSS quantifies the deviation of a value at a given time ($t_0$) from the linear interpolation
line calculated from the preceding and succeeding time series data points (Eq. 1).

$$\text{TSS}(t_0) = \frac{|(X(t_1)-X(t_{-1}))\times t_0-(t_1-t_{-1})\times X(t_0)-(X(t_1)-X(t_{-1}))\times t_{-1}+(t_1-t_{-1})\times X(t_{-1})|}{\sqrt{(X(t_1)-X(t_{-1}))^2+(t_1-t_{-1})^2}} \quad (1)$$

We analysed three adjacent time series data points, $X(t_1)$, $X(t_0)$, and $X(t_{-1})$, obtained at the following ($t_1$), current ($t_0$), and
the previous ($t_{-1}$) times, respectively. To ensure a fair comparison, we calculated the cumulative TSS based on the same length
of time series. The TSS represents the deviation of a value at a given point in time from the linear interpolation line, and in
this study, higher TSS values indicate greater variability over time.

The MAE metric is employed in this study to evaluate the accuracy of the retrieval methods by measuring the average
absolute difference between the predicted and actual values. Following the approach proposed by (Zhou et al., 2015), the MAE
is calculated as the discrepancy between the reference and the retrieved series. We used the MAE as a metric to assess the

similarity between the retrieved and the reference series (Eq (2)). The process of generating the reference time series will be discussed in Section 3.2.

$$MAE = \sqrt[2]{\frac{\sum_{i=1}^{N}(retrieved(i)-reference(i))^2}{N}} \tag{2}$$

Furthermore, performance analyses indicate that the retrievals from the main algorithm without clouds and aerosols provide the highest quality and accuracy (Pu et al., 2020). Therefore, we used a retrieval index (RI), which represents the percentage of pixels with high-quality retrievals (Xu et al., 2018;Yan et al., 2018;Yan et al., 2021a), as an indicator of the uncertainty of the LAI/FPAR retrievals (Eq. 3). This *RI* serves as an additional measure to assess the uncertainty associated with LAI/FPAR retrievals.

$$RI = \frac{Number\ of\ high-quality\ pixels}{Number\ of\ all\ pixels} \tag{3}$$

**2.6 Calculation of LAI/FPAR trends**

    Trends in annual average SILAI/FPAR CDR (2001 to 2022) are evaluated by the Mann–Kendall (MK) test. The MK test is a non-parametric statistical test commonly used for climate diagnostics and prediction. It enables the detection of monotonic trends in time series data, helping to determine if significant trends exist (Hamed and Rao, 1998). The MK test is employed as
follows:

$$S = \sum_{i=1}^{n-1}\sum_{j=i+1}^{n} sgn(x_j - x_i) \tag{4}$$

$$Var(S) = \frac{n(n-1)(2n+5)-\sum_{i=1}^{m} t_i(t_i-1)(2t_i+5)}{18} \tag{5}$$

$$Z_s = \begin{cases} \frac{S-1}{\sqrt{Var(S)}}, & \text{if } S > 0 \\ 0, & \text{if } S = 0 \\ \frac{S+1}{\sqrt{Var(S)}}, & \text{if } S < 0 \end{cases} \tag{6}$$

    Equation 4 calculates the sum (*S*) of step function values, which represent the differences between values at different
points ($x_j$ and $x_i$) in the time series. The variables *n* and *m* denote the number of data points and the number of tied groups (recurring data sets), respectively. Next in Equation 5, the variance (*Var(S)*) is calculated by assessing the magnitude of *S* to evaluate the statistical significance of the detected trends. Where the $t_i$ is the number of the ties (the number of repeats in the extent *i*). Finally, we calculated the test statistic $Z_s$ (Equation 6). When $|Z_s| > Z_{1-\alpha/2}$, the null hypothesis (i.e., no trend) is rejected and the $\alpha$ is a special significance level. Here, we use the significance level of $\alpha = 0.05$ and the $Z_{1-\alpha/2} = 1.96$. Thus,
the trends with *P ≤ 0.05* are considered to be statistically significant in this study.

Data

# 3 Methodology



**Figure 2.** Schematic flowchart of the generation of the SI LAI/FPAR CDR.



Our methodology includes four key steps as shown in Fig. 2: 1) Filtering low-quality observations based on QA values;
2) Consolidating the Filtered Terra/Aqua/VIIRS LAI/FPAR into Filtered SI LAI/FPAR timeseries; 3) Gap filling the missing values using a spatial-temporal tensor completion model; 4) Generating the SI LAI/FPAR CDR in different projections, spatial and temporal resolutions. The details are described in the following subsections.

### 3.1 Step1: Filtering the Low-Quality Observations based on QA Values

The FparLai_QC and FparExtra_QC layers within the MODIS&VIIRS LAI/FPAR products provide information about
the quality of LAI/FPAR, which include algorithm path, cloud and aerosol contaminations, and other factors that lower the reliability of retrievals. Best-quality, high-precision retrievals are obtained from the main algorithm. With a high probability the main algorithm fails in the case of cloud and/or snow-contaminated pixels, or pixels with a high aerosol load (Yan et al., 2021a). As depicted in Table 1, the first step eliminates LAI/FPAR values retrieved by the backup algorithm as well as suspicious values from the main algorithm, which could be affected by clouds, aerosols, and cloud shadows. Note that the
quality flags of the Terra-MODIS and Aqua-MODIS products are consistent, while VIIRS slightly differ from the two MODIS QA (e.g., the absence of MODLAND_QC for VIIRS). All three LAI/FPAR products have the same algorithm path and atmospheric conditions. After removing suspicious and low-quality values we get a Filtered Terra/Aqua/VIIRS LAI/FPAR data set.

**Table. 1** The quality flags used in the step1.

| | Terra | Aqua | VIIRS |
|---|---|---|---|
| MODLAND_QC | Good quality (main algorithm) | Good quality (main algorithm) | × |
| Retrieval Algorithm Path | main algorithm | main algorithm | main algorithm |
| Cloud State | Clear or assume clear | Clear or assume clear | Confident clear or probably clear |
| cloud shadow | No cloud shadow detected | No cloud shadow detected | No cloud shadow |
| internal cloud mask | No clouds | No clouds | × |
| Cirrus | No cirrus detected | No cirrus detected | No |
| LandSea Path | Land | Land | × |
| Fill value | No | No | No |

**3.2 Step2: Consolidating the Filtered Terra/Aqua/VIIRS LAI/FPAR into Filtered SI LAI/FPAR Timeseries**

After completing the first step, we obtained three Filtered LAI/FPAR timeseries of different spatial coverage. We consolidated the three timeseries into a Filtered SI LAI/FPAR timeseries as follows. First, a fill value assigned to a pixel, if



there are no high-quality LAI/FPARs from any sensor during the compositing period. Second, if there is only one pair of high-quality LAI and FPAR for the pixel, it is taken as the CDR value with accompanying quality flag set to 1. Finally, if several

high-quality retrievals are available for a given pixel during the compositing period, their average is the CDR value. The corresponding quality flag is set to 1 in this case. We generated a 23-year (2000-2022) time series of the Filtered SI LAI/FPAR using this procedure. As (Zhou et al., 2015) argued this timeseries can be used as a reference to estimate an absolute reconstruction error.

### 3.3 Step3: Gap Filling the Missing Values Using a Spatial-temporal Tensor Completion Model

Our next step is gap filling, i.e., replacing fill values with estimates of LAI and FPAR. We used a ST-Tensor completion model as our spatial-temporal gap filling method. This method is particularly well suited for correcting remote sensing images due to its ability to capture intrinsic multidimensional correlations. This technique has proven successful in many applications such as hyperspectral image recovery and reconstruction of missing data in remote sensing images (He et al., 2017;Zheng et al., 2019;Zhang et al., 2019). The ST-Tensor completion model used in this study can be broken down into the following four

steps.

1) Tensor rearrangement: The LAI/FPAR timeseries have three distinct features. First, they possess spatial neighbourhood similarity based on the first law of geography (Goodchild, 2009). Second, they have temporal neighbourhood correlation, assuming vegetation growth is continuous and smooth (Cong et al., 2012). Finally, they exhibit periodic temporal similarity because vegetation growth varies periodically (Whitt and Ulaby, 1994). While the original tensor form adequately captures

spatial neighbourhood similarity and temporal neighbourhood correlation, it does not directly express periodic temporal similarity. Therefore, it is necessary to transform the original tensor into a new tensor that includes all three features. As shown in step 3 of Fig. 2, this study transforms the one-dimensional multi-year time series for each pixel into a two-dimensional matrix. In this matrix, each row represents a one-year time series. At the same time, the two-dimensional spatial image is transformed into a one-dimensional vector. The following equation illustrates this transformation:

$$X_{m*m*T} \rightarrow Y_{m^2*ny*nd} \tag{7}$$

where $X$ is the original tensor, $Y$ is the transformed tensor, $m$ denotes the spatial length of the original tensor, and $T$ denotes the total number of observations in the entire timeseries, which is basically equal to the number of years ($ny$) multiplied by the number of observations in a year ($nd$).

2) Iteration updates the weight values and gap fills the missing values: The current third-order tensor can be decomposed

into three matrices by three different modes. The tensor rank is defined in different ways and is mainly determined by the correlation and similarity of the elements in the different domains of the tensor. A smaller tensor rank indicates a higher similarity between the values of the elements of the tensor, which also implies that the missing values are filled consistently across all three dimensions, resulting in the best gap filling. The tensor rank can be defined by considering the three expansion matrices' rank order. Therefore, the data filling process in this study can be represented by the following equation, which aims

to optimize problem solving:

$$min \sum_{i=1}^{3} w_n \ rank(Y_n) \tag{8}$$

where $w_n$ is the weight corresponding to $Y_n$ that is always non-negative and satisfies $\sum_{i=1}^{3} w_n = 1$. In the process of solving equation (8), it is first necessary to iteratively determine the weight values. We initialize the weights of the three expansion matrices as equal ($w_1 = w_2 = w_3 = 1/3$) and then update them iteratively using the method based on singular value decomposition introduced by (Chu et al., 2021). Once the weight values are obtained, Eq. (8) can be solved efficiently using the algorithm proposed by (Ji et al., 2017). This algorithm uses a logarithmic operator, which better approximates the tensor rank than the classical kernel parameterization, leading to higher accuracy (Ji et al., 2018).

3) Iterate L1 trend filtering: After the ST-Tensor completion process, although a gap-free LAI/FPAR timeseries can be obtained, some residual noise may still exist due to the uncertainties in Flag data. Thus, we employ an iterative L1 trend filtering method, known for its flexibility in denoising one-dimensional timeseries data by regularizing the residual and smoothing terms (Chu et al., 2021;Chu et al., 2022;Eilers, 2003). In this method, we denote the noisy time series as *y* and the fitted series as *z*. The objective of the L1 trend filtering method is to balance two conflicting objectives: (a) fidelity to the original series and (b) smoothness of the filtered series. This is achieved by optimizing the following objective function (Eq. 9).

$$Q = \frac{1}{2} \|y - z\|_2^2 + \lambda \|D_z\|_1 \tag{9}$$

where $D_z$ represents the second-order difference matrix and $\lambda$ is the regularization parameter that balances the fidelity and smoothness terms.

In practice, this method is well suited to preserving the detailed characteristics of turning points, thanks to the L1 parametric constraint. Furthermore, due to the completion of the ST-Tensor process, the remaining noise can be assumed to have a negative bias. In our approach, data with Flag = 0 is considered as noise, while Flag = 1 is considered as almost noiseless. The iterative process is as follows: In the first and second iterations, the L1 trend filter is applied to smooth the LAI/FPAR time series, preserving the good data while replacing only the noisy values that fall below the smoothed series. Subsequent iterations then replace all the noise. By iteratively repeating this process, we achieve a good balance between noise reduction and the preservation of good data. At the end of the filtering step, the results are both gap- and noise-free.

4) Reshape the tensor: Following the reverse process of rearrangement in step 1), the filled and filtered LAI/FPAR tensor is returned to its original form.

### 3.4 Step4: Generating the SI LAI/FPAR CDR at Different Projections, Spatial and Temporal Resolutions

The final step aims to generate CDRs at different spatial and temporal resolutions projected on WGS1984 and sinusoidal grids. First, we calculated the bimonthly LAI/FPAR using a weighted averaging method. The spatial resolution of the data was then adjusted to 5 km by nearest neighbour interpolation. Following this step, the projection was transformed from sinusoidal to WGS1984 and the data were further interpolated from a spatial resolution of 500m down to 0.05 degrees using the block average method. As a result of these transformations, we obtained six different versions of the SI LAI/FPAR CDR, as shown



in Table 2. Finally, the SI LAI/FPAR CDR with 500m spatial resolution were uploaded to Google Earth Engine (GEE) for users to mix and match with other datasets and the ease of using this in GEE, all datasets were reprojected to WGS1984 using

the gdalwarp function with a crs of EPSG:4326 for ease of ingestion. The other four versions of SI LAI/FPAR CDR can be found in Zenodo (detailed in the section Data Availability).

**Table 2.** Projections and spatial/temporal resolutions of SI LAI/FPAR CDR.

| Resolutions' ID | Projection | Spatial Resolution | Temporal Resolution | Dimensions | Repository |
|---|---|---|---|---|---|
| 500m_8day | WGS1984 | 500m | 8 days | 43200 x 86400 rows/columns | GEE |
| 500m_bimonth | WGS1984 | 500m | Half month | 43200 x 86400 rows/columns | GEE |
| 5km_8day | Sinusoidal | 5km | 8 days | 4320 x 8640 rows/columns | Zenodo |
| 5km_bimonth | Sinusoidal | 5km | Half month | 4320 x 8640 rows/columns | Zenodo |
| 0.05degree_8day | WGS1984 | 0.05 degree | 8 days | 3600 x 7200 rows/columns | Zenodo |
| 0.05degree_bimonth | WGS1984 | 0.05 degree | Half month | 3600 x 7200 rows/columns | Zenodo |

## 4 Results

### 4.1 Evaluation for the Generating Steps of SI LAI/FPAR CDR over the Amazon Forest

The assessment of vegetation dynamics in the Amazon rainforest, including seasonal changes, is challenging due to persistent cloud cover that interferes with optical remote sensing observations. Therefore, during the evaluation process of the SI LAI/FPAR, each step was assessed using the Amazon Forest region (Fig. 2) as a representative study area. In the first step, considering the significant presence of EBF (50.26%) in the Amazon Forest region, this part compares the LAI/FPAR timeseries from Terra/Aqua/VIIRS before and after filtering the low-quality observations. This filtering process has reduced

the magnitude of the LAI variation, leaving their values in the range of 5-6 (Fig. 3), as expected (Samanta et al., 2012a;Samanta et al., 2012b). Filtered FPAR values vary around its inter-annual average of 0.85 (Fig. S1). Furthermore, we observed a consistent intra-annual variation in LAI/FPAR, indicating a clear seasonal dynamic in the EBF.



**Figure 3.** The temporal comparisons between the original Terra/Aqua/VIIRS LAI and Filtered Terra/Aqua/VIIRS LAI for the EBF of Amazon Forest region. The panel (a)~(c) represent Terra/Aqua/VIIRS, respectively. And the blue and purple lines mean original and filtered LAI.

Fig. 4 illustrates percentages of high-quality retrievals from single sensors and after step 2 for selected Amazon Forest region (zoom-in case in Fig. 1) for the 2013 to 2022 overlap period. Table 3 summarizes the changes after the implementation of step 2 for the whole and overlap periods. For the overlap period the percentages increase from 41.09% (Terra, 14.37%+7.29%+6.98%+12.45%), 29.55% (Aqua), and 30.90% (VIIRS) to 58.50% (100%-41.50%) in Filtered SI timeseries. If the time interval is extended to the whole dataset record period (2000 to 2022), the percentages increase from 41.68%, 26.73%, and 14.85% to 54.07%. Consequently, the percentage of pixels that need to be gap filled reaches 45.93%. Table 3 also shows a low frequency of high-quality retrievals available simultaneously from all sensors: 12.45% for the overlap period and 6.02% for the entire acquisition period. Furthermore, Fig. 4 also illustrates that the percentage varies with biome type: it





takes values of 46.30% for EBF (northwest region) 67.51% for savannas (southeast region). Fig. 5 highlights the discontinuity of the spatial-temporal distribution of the quality flag generated by the data consolidation algorithm (Section. 3.2). For example, the RI for DOY 001 in 2014 (rain season) is 52.66%, while for DOY 193 in 2014 (dry season) it is 76.63%. This lack of a clear pattern in the spatial-temporal distribution of the pixels to be gap filled places significant demands on the subsequent filling process.

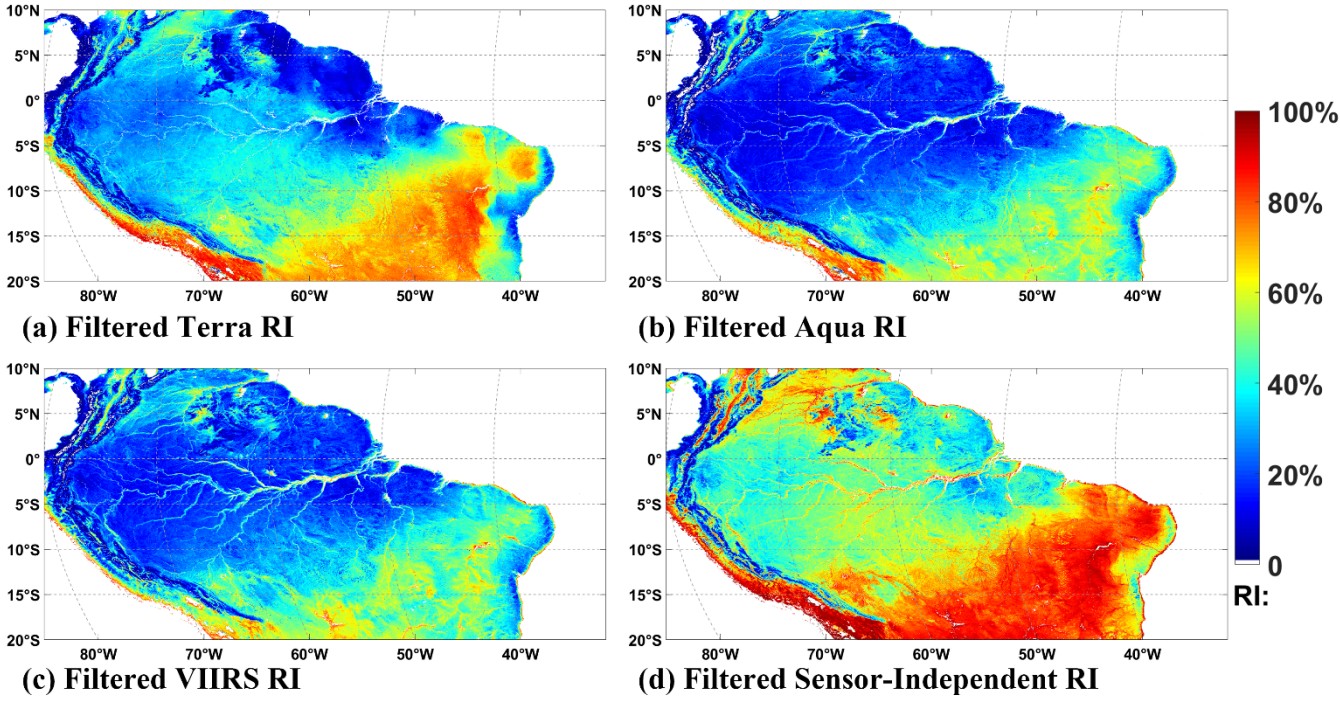

**Figure 4.** The spatial distribution of filtered Terra (a), Aqua (b), VIIRS (c), SI (d) RI in the selected Amazon Forest region (zoom-in case in Fig. 1) in overlap period (2013 to 2022).

**Table 3.** Percentage of pixels with high-quality retrievals from single sensors and their combinations for whole (2000 to 2022) and overlap (2013 to 2022) periods. Each row sums to 100%.

| | No data | Terra Only | Aqua Only | VIIRS Only | Terra & Aqua | Terra & VIIRS | Aqua & VIIRS | All Sensors |
|---|---|---|---|---|---|---|---|---|
| **2000 ~ 2022** | 45.93% | 20.35% | 6.91% | 3.62% | 11.95% | 3.36% | 1.85% | 6.02% |
| **2013 ~ 2022** | 41.50% | 14.37% | 5.94% | 7.60% | 7.29% | 6.98% | 3.87% | 12.45% |

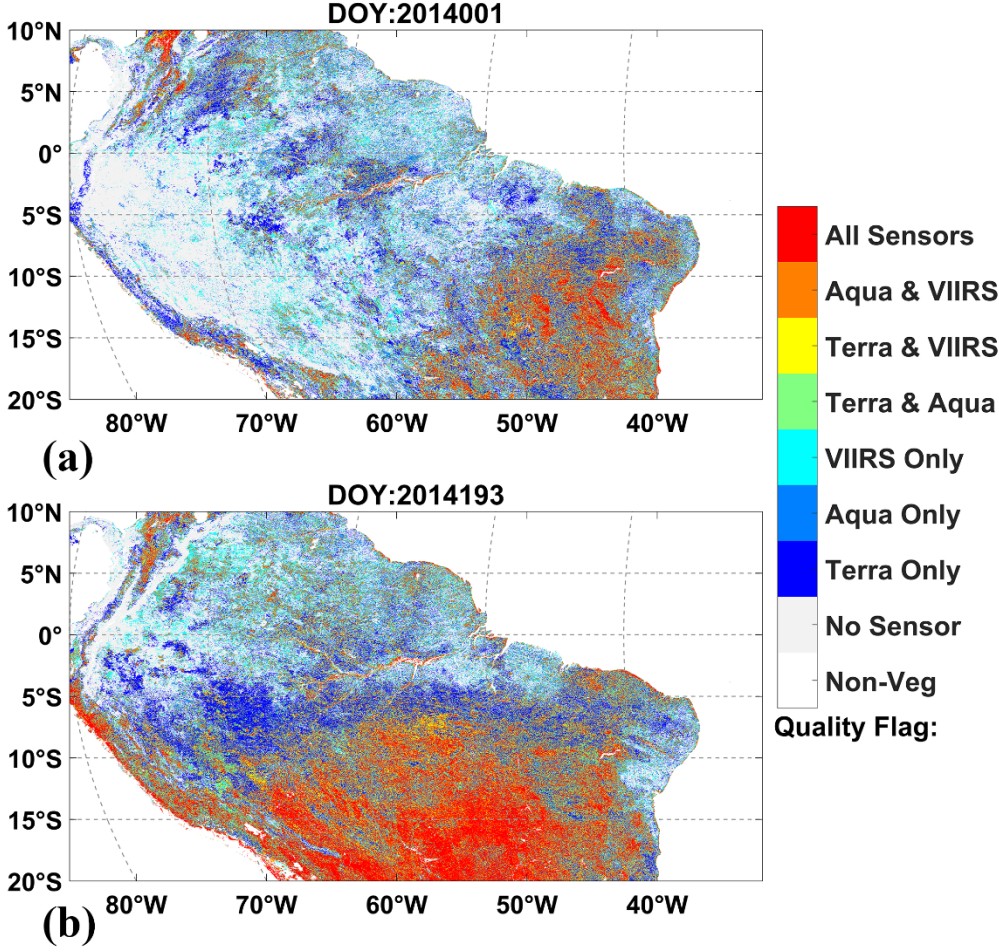

**Figure 5.** Spatial distribution of the quality flag of the data consolidation algorithm (Sect. 3.2). Panels a and b represent DOYs 1 and 193. Flag value of 0 is grey coded, while other colours correspond to quality flag=1 for various combinations of the sensors. The color corresponding to Flag=0 represents the No-Sensor category, while the other colors in the figure represent Flag=1. Replace "No Sensor" with "Quality flag=0" and "Non-Veg" with "Non-vegetated land". Delete "Quality Flag" from color bar.

In step3, we replaced fill values with LAI/FPAR estimates produced by the ST-Tensor completion model. The goal of this section is to assess the performance of this technique in terms of differences between the gap filled LAI/FPAR and the high-quality Filtered SI LAI/FPAR consolidated in Step 2. The latter is taken here as the reference data set. To address this issue, we removed 5% of the high-quality Filtered SI LAI/FPAR with flag=1. There were 1,574,181,910 such pixels in our Amazon region, covering the period from 2000 to 2022. First, we estimated LAI/FPAR values for these pixels using the ST-Tensor completion model applied to the remaining 95% of the reference LAI/FPAR. The estimates were then compared with the removed values. This analysis was performed for each biome type found in the Amazon region.



**Figure 6.** Comparisons of Filtered SI LAI and ST-Tensor SI LAI from 2000 to 2022 year for 5% pixels that Flag =1 in the selected Amazon Forest region (zoom-in case in Fig. 1). The dotted symbols are color-coded DOY; black and blue lines represent the diagonals and linear fit 340 lines, respectively. Panels (a)-(g) represent the seven biome types from B1-B7. B8 is not shown due to its very small number in the Amazon Forest region. Panel (h) represents all selected pixels. The N means the whole pixels number from 2000 to 2022 for different biome types.

First, Figs 6 and S2 clearly illustrate a very good consistency between reference and estimated LAI/FPAR values in terms of $R^2$: its value exceeds 0.90 for both LAI/FPAR and all biome types, suggesting a strong relationship and good prospects for our gap filling approach. Specially, both LAI and FPAR have $R^2$ values above 0.99 for most biome types with RMSE below

0.14 (LAI) and 0.02 (FPAR) for all biome types. When we merged all biome types, the $R^2$ become 0.95 (LAI) and 0.98 (FPAR) with RMSE values of 0.08 and 0.01, respectively. Second, both ST-Tensor generated LAI and FPAR show a slight underestimation of the reference values and the slight lower $R^2$ (0.92) for EBF, but still demonstrating a high degree of consistency with RMSE below 0.13 and 0.01 for LAI and FPAR, respectively. Finally, we observe the significant seasonality of EBF, with LAI decreasing from January to June and increasing thereafter. This phenomenon matches the phenology shown

in Step 1. In summary, these results suggest an excellent performance of the ST-Tensor completion model in gap filling the missing LAI/FPAR values, resulting in a good agreement with the original high-quality SI LAI/FPAR.

## 4.2 Intercomparison at the Global scale

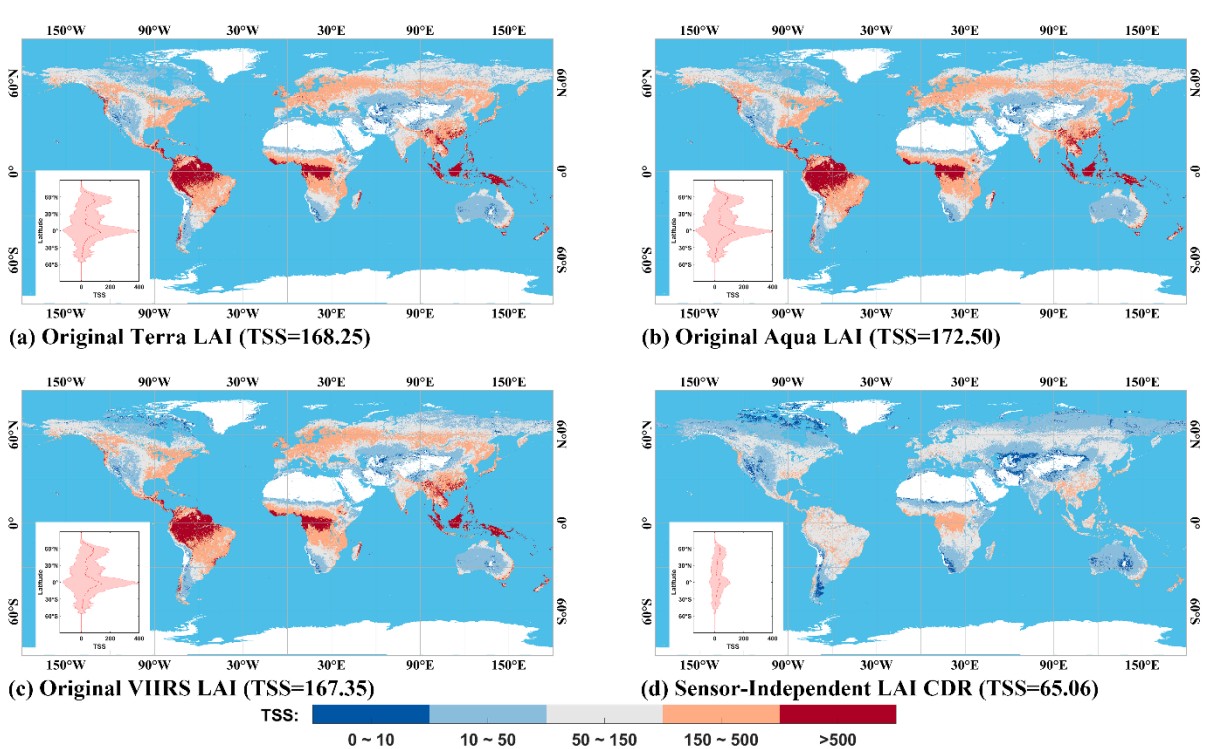

(a) Original Terra LAI (TSS=168.25)  (b) Original Aqua LAI (TSS=172.50)

(c) Original VIIRS LAI (TSS=167.35)  (d) Sensor-Independent LAI CDR (TSS=65.06)

TSS: 0 ~ 10  10 ~ 50  50 ~ 150  150 ~ 500  >500

**Figure 7.** The global distribution of LAI TSS in each 0.05 degree× 0.05 degree grid from 2013 to 2022. Panel (a) –(c) displays the TSS of

original Terra/Aqua/VIIRS LAI, respectively. While panel (d) shows the TSS of SI LAI CDR. A WGS1984 projection is used here and the temporal resolution is 8 days. Panels right represent latitudinal transects (0.05 degree interval) of TSS values for LAI. Red lines and shadows represent the mean values and standard deviations of LAI TSS of the 0.05 degree latitude zone.

Here we evaluate the SI LAI/FPAR CDR as well as the LAI/FPAR derived from the original Terra/Aqua/VIIRS data using the TSS and MAE metrics. The results suggest significant improvments in the SI LAI/FPAR CDR compared to the

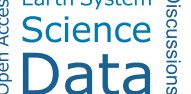

original Terra/Aqua/VIIRS LAI/FPAR data sets. As shown in Figs. 7 – 8 and Figs. S3-S4, SI LAI/FPAR CDR (8-day temporal

resolution, 0.05 degree spatial resolution and WGS84 projection) have a higher number of "blue" pixels and lower number of

"red" pixels, indicating lower TSS and MAE values. Specifically, the TSS (Fig. 7) of SI LAI CDR (65.06) is much lower than

original Terra LAI (168.25), original Aqua LAI (172.50), and original VIIRS LAI (167.35). Similarly, the TSS (Fig. S3) of SI

FPAR CDR (12.67) is the lowest compared to the original Terra FPAR (39.68), original Aqua FPAR (41.18), and original

VIIRS FPAR (37.94). These indicate that the SI LAI/FPAR CDR has reduced volatility and noise compared to the original

versions. The MAE (Fig. 8) shows a similar trend as TSS, with the MAE decreasing from 0.80 (original Terra LAI), 0.79

(original Aqua LAI), and 0.75 (original VIIRS LAI) to 0.39 (SI LAI CDR). And the MAE decreases from 0.16 for the original

Terra FPAR, 0.17 for the original Aqua FPAR and 0.16 for original VIIRS FPAR to 0.10 for SI FPAR CDR (Fig. S4). This

suggests that the SI LAI/FPAR CDR is more closely related to a high-quality LAI/FPAR reference than the original

Terra/Aqua/VIIRS LAI/FPAR. In addition, the variation of TSS/MAE with latitude confirms these improvements, as the SI

LAI/FPAR CDR shows smaller mean values and standard deviation of TSS/MAE compared to the original Terra/Aqua/VIIRS

LAI/FPAR.

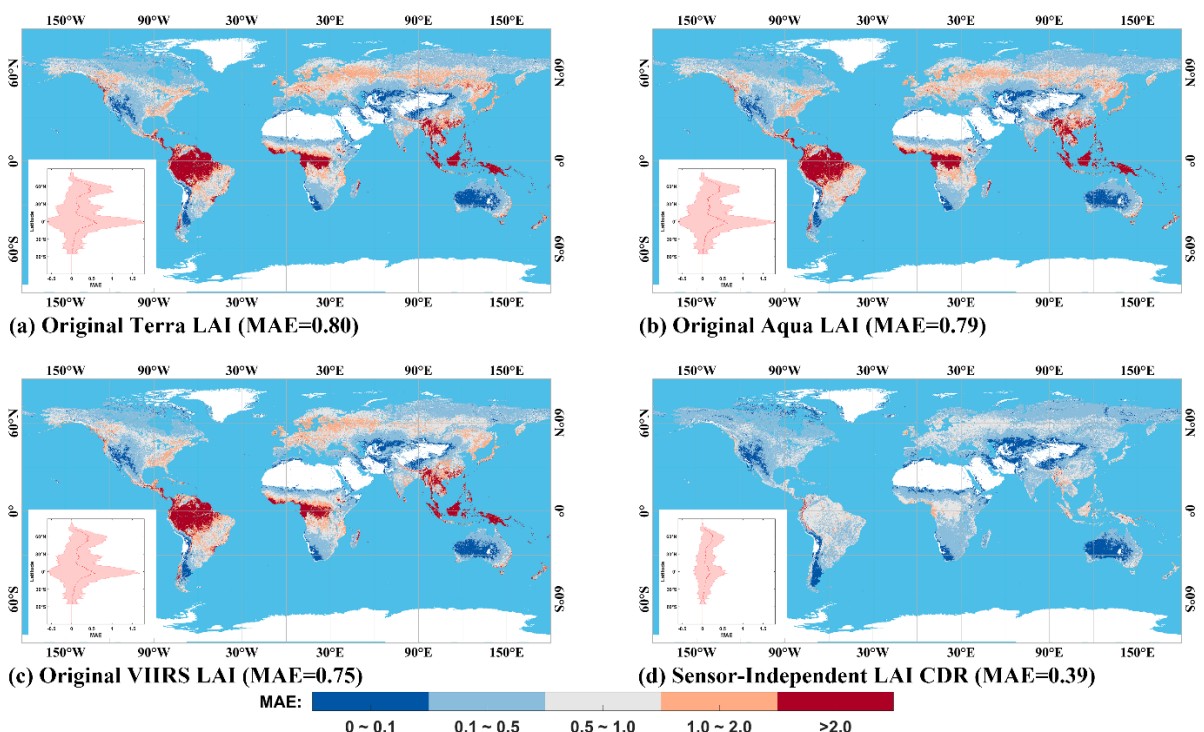

**Figure 8.** Same as Fig. 7 but the metric is MAE. The global distribution of LAI MAE in each 0.05 degree× 0.05 degree grid from 2013 to
375    2022.

Evidently, these improvements are particularly noticeable in the selected Amazon Forest region, where both the TSS and

MAE of SI LAI/FPAR CDR show significant decreasing trend compared to the original Terra/Aqua/VIIRS LAI/FPAR (Figs.

S5 – S8). The magnitude decrease of TSS for SILAI/FPAR CDR is above 300/60, and the magnitude decrease of MAE is





above 1.5/0.2, which exceeds other global regions. This implies a significant improvement in the SI LAI/FPAR CDR,
especially when dealing with large amounts of missing data. Furthermore, similar improvements are observed when the
temporal resolution is bimonthly (Figs. S9 – S12). Overall, these results highlight the significant improvement achieved with
the SI LAI/FPAR CDR over the original Terra/Aqua/VIIRS LAI/FPAR, demonstrating its enhanced performance and accuracy.

**4.3 Validation Using Ground LAI/FPAR Measurements**



**Figure 9.** Comparisons of original Terra/Aqua/VIIRS LAI and SI LAI CDR with ground GBOV LAI. The bin for X and Y axis is 0.1 and
the color means the number of pixels in this bin.



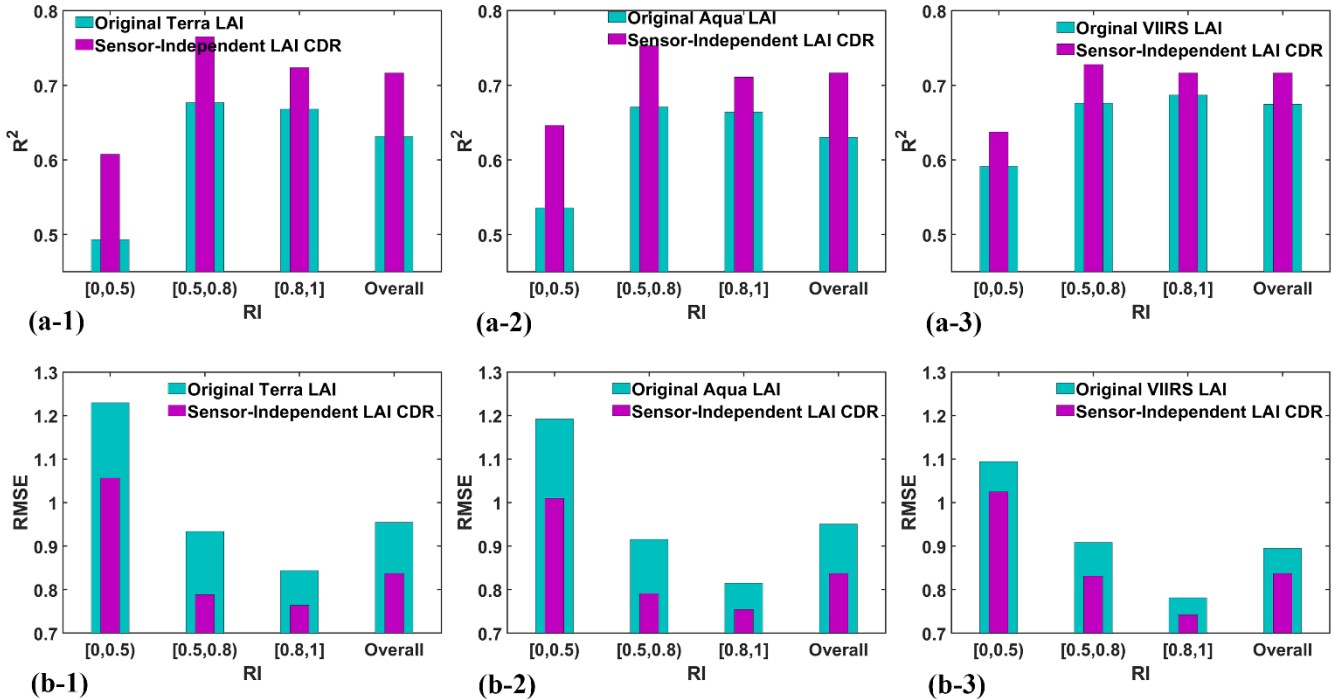

**Figure 10.** The $R^2$ and RMSE between original Terra/Aqua/VIIRS LAI and SILAI CDR and GBOV LAI in different RI ranges.

Fig 9 shows comparisons of the SI LAI CDR and original Terra/Aqua/VIIRS LAI with the GBOV ground data. The SI
LAI CDR shows highest accuracy, as evidenced by $R^2 = 0.72$ and RMSE = 0.84. Followed by the original VIIRS LAI ($R^2 =$
0.67 and RMSE = 0.89) and then the original Terra/Aqua LAI ($R^2 = 0.63$ and RMSE = 0.95 for both Terra/Aqua). Similar
results are observed in the case of FPAR, where the Sensor Independent FPAR CDR outperforms the original
Terra/Aqua/VIIRS FPAR. The $R^2$ values increased from original Terra/Aqua/VIIRS FPAR by 0.74/0.72/0.76 to 0.79 (Sensor
Independent FPAR CDR), while the RMSE decrease from 0.17/0.18/0.16 to 0.15 (Fig. S13). It is important to note that all
datasets show a tendency to overestimate low LAI/FPAR values and underestimate high FPAR values. However, the Sensor
Independent LAI/FPAR CDR shows a closer fit to the 1:1 line compared to the original Terra/Aqua/VIIRS LAI/FPAR. In
addition, the SI LAI/FPAR CDR shows significant improvements across all RI ranges. The $R^2$ values increased by
approximately 0.02 to 0.21 from the original Terra/Aqua/VIIRS LAI/FPAR to SI LAI/FPAR CDR, while the RMSE values
decrease by approximately 0.01 to 0.22 (Fig. 10 and Fig. S14). Especially the RI in the range of [0, 0.5], the enhancements
were the most significant among all RI ranges for both Terra/Aqua/VIIRS, which indicates that the Sensor Independent
LAI/FPAR CDR represents a significant improvement over the LAI/FPAR obtained using the backup algorithm.



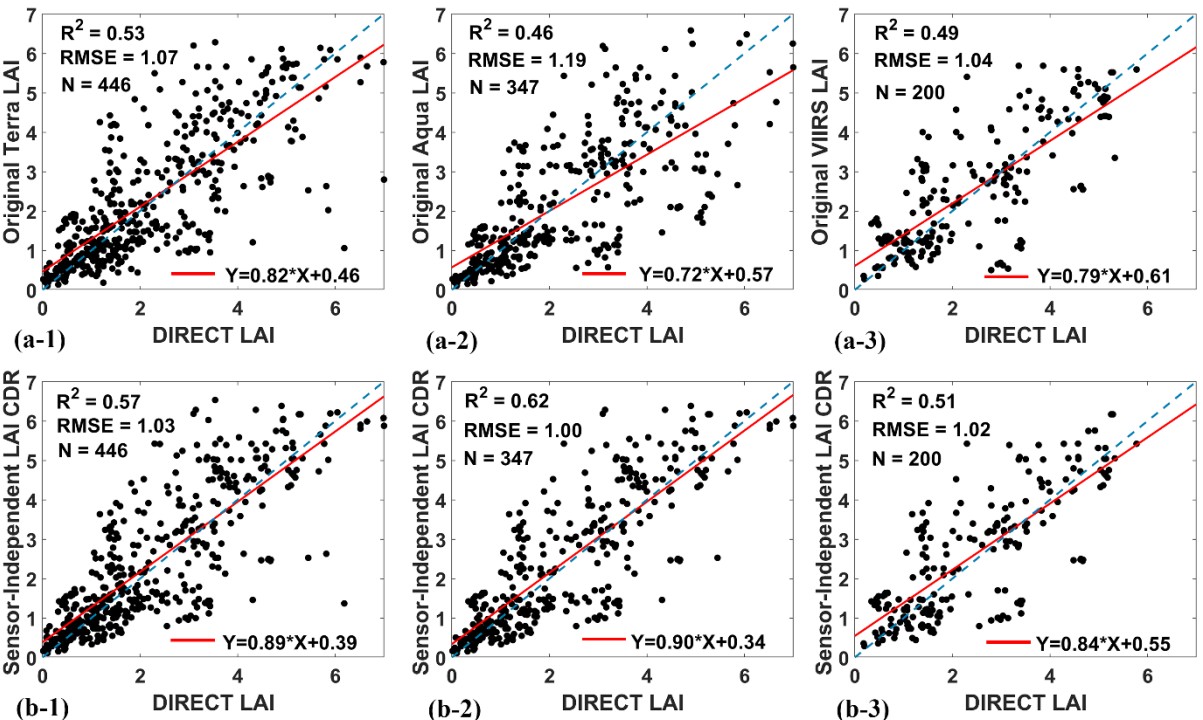

**Figure 11.** Comparisons of original Terra/Aqua/VIIRS LAI and SI LAI CDR with ground DIRECT2.1 LAI measurements. The numbers corresponding to N in the top plot are the number of all DIRECT measurements for the period covered by Terra/Aqua/VIIRS, respectively, and the N in the bottle plot corresponds to N in the top plots, respectively.

Improvements were also observed based on comparisons with the DIRECT ground truth data. Compared to the original Terra/Aqua/VIIRS LAI, the $R^2$ values of Sensor Independent LAI CDR increased by 0.04/0.14/0.02, while the RMSE values showed a decrease by 0.04/0.19/0.02 (Fig. 11). Similarly, there is an increasing trend in $R^2$ of 0.09/0.05/0.06 for FPAR, accompanied by a decreasing trend in RMSE of 0.02/0.01/0.02 (Fig. S15). The improvement from original Aqua LAI to Sensor Independent LAI CDR was the most pronounced in LAI, with $R^2$ increasing from 0.46 to 0.62 and RMSE decreasing from 1.19 to 1.00. The improvement from original Terra FPAR to Sensor Independent FPAR CDR was the most pronounced in FPAR, with $R^2$ increasing from 0.71 to 0.82 and RMSE decreasing from 0.13 to 0.11. The improvement is also reflected in the fact that the Sensor-Independent LAI/FPAR scatters vs. DIRECT LAI/FPAR are closer to the 1:1 line.



## 4.4 Revisit the Greening Earth



### (a) Sensor-Independent LAI CDR

### (b) Sensor-Independent FPAR CDR

Trend in annual average LAI/FPAR from 2001 to 2022  (%/decade)

**Figure 12.** Map of trends in annual average Sensor-Independent LAI/FPAR for 2001–2022. Statistically significant trends (Mann–Kendall test, P≤0.05) are color-coded. Grey areas show vegetated land with statistically insignificant trends. White areas depict barren lands, permanent ice-covered areas, permanent wetlands, and built-up areas. Blue areas represent water.





Previous studies have shown a significant greening trend in global leaf area (Zhang et al., 2017;Chen et al., 2019;Cortés
et al., 2021). However, loss of sensor calibration, atmospheric contamination of the vegetation signal influenced the previous
studies. Thus, this study aims to provide further insight into 'Greening Earth' by analysing the high-quality SI LAI/FPAR
CDR. The results obtained from this study provide a clear indication of the greening trend of vegetation. Specifically, according
to SI LAI CDR, almost one-third (30.71%) of the vegetated areas are greening and only 6.67% of the vegetated areas show a
browning trend (Fig. 12a). These findings are also supported by the results of the SI FPAR CDR, where about 31.26% of the
vegetated area is greening and only 6.50% is browning (Fig. 12b). In terms of overall global vegetation status, the analysis
suggests a significant increase in greenness from 2001 to 2022. The LAI shows a significant increasing trend of 2.33% per
decade and the FPAR also shows a significant increasing trend of 1.93% per decade, which can be translated to a constant net
increase in leaf area. Focus on individual regions, SI LAI/FPAR CDR also highlights the significant contribution of China and
India to the global greening trend. These regions show significant increases in vegetation greenness, supporting the overall
global trend observed in the analysis. While there are scattered browning trend in the high latitudes of the northern hemisphere,
south Africa and Amazon Forest region.

**5 Discussion**

Improving the quality of LAI/FPAR timeseries is crucial to ensure the reliability of vegetation studies. Therefore, it is
necessary to explore various techniques of post-processing satellite-derived LAI/FPAR. In the operational data processing, the
main RT-based algorithm produces best quality and high precision parameters. With a high probability the main algorithm
fails in the case of cloud and/or snow contaminated pixels, or pixels with a high aerosol load. When this happens, the backup
method is utilized. This causes high frequency noise in the LAI/FPAR time series, which indirectly leads to time series
instability and absolute accuracy problems. Therefore, the first step in the developing SI LAI/FPAR CDR production process
was to eliminate poor quality retrievals to ensure that subsequent steps such as gap filling are not affected by noise. As a result,
the first step filtering principle is also relatively more stringent. Previous studies on the reconstruction of MODIS&VIIRS LAI
time series often only filter the backup algorithm (Huang et al., 2021) but this study also filters the main algorithm that have
been affected by clouds and aerosols. The findings after step 1 are consistent with previous studies investigating the dynamics
of vegetation phenology in tropical rainforest (Myneni et al., 2007;Samanta et al., 2012a;Hashimoto et al., 2021;Sun et al.,
2022), which also indirectly proves the necessity of this step.
The current MODIS&VIIRS LAI/FPAR retrieval algorithm considers the effect of SZA on LAI, but also FPAR depends
on SZA (Knyazikhin et al., 1998a;Pu et al., 2020). The newly proposed SI LAI/FPAR CDR differs from the previously
reconstructed sensor-dependent time series by its SI feature, which includes both morning and afternoon high-quality retrievals.
By accounting for the effects of satellite transit times at different times of the day, particularly in relation to SZA variations,
the SI LAI/FPAR CDR avoids potential systematic biases. Thus, the SI property allows FPAR to be unaffected by a single



observation. In addition, the process of consolidating the original Terra/Aqua/VIIRS LAI/FPAR into a SI LAI/FPAR also
provides as much information as possible for the subsequent gap filling step (Ganguly et al., 2008;Xiao et al., 2014).

   In recent years, numerous spatial-temporal reconstruction methods have been proposed from different perspectives.
However, most of these methods are based on empirical filtering and function fitting approaches, neglecting the use of prior
information or statistical properties. The ST-Tensor model used in this study differs from previous models in that it considers
the strong correlation between temporal and spatial scales (Chu et al., 2021). In addition, the ST-Tensor model considers the
consistency of variability across years, effectively exploiting the internal correlation of the LAI/FPAR (Ji et al., 2018;Li et al.,
2019). As a result, the gap filled LAI/FPAR shows a high degree of consistency with the original high-quality data, thus
preserving the integrity of the original measurements. This reduction in the frequency of noise within the LAI/FPAR time
series is highly beneficial for phenology studies and agricultural management. Additionally, the L1 trend iteration step of the
ST-Tensor model ensures that certain anomalous observations, such as forest degradation due to fire, are not inadvertently
smoothed by the algorithm. This capability contributes to the reliability and accuracy of the model.

   This study establishes the accuracy of the SI LAI/FPAR CDR through direct ground validation, which includes rigorous
evaluation against ground measurements and related metrics. The results of this validation process demonstrate the reliability
of the product. Furthermore, we also compared the above analysing results (section 4.4) reported by a recent study (Chen et
al., 2019;Zhang et al., 2017), both greening and browning trends are consistent. These results confirm the previous
understanding of 'Greening Earth' and indirectly prove the reliability of SI LAI/FPAR CDR. The SI LAI/FPAR CDR analysis
provides a valuable tool for monitoring and understanding the dynamics of global vegetation change. The consistency of the
study's results with those of previous research adds to the robustness and credibility of the product. The provision of multiple
spatial and temporal resolution versions of the SI LAI/FPAR CDR in this study greatly enhances the ability to study global
and local vegetation change and climate dynamics.

**6 Conclusions**

   This study developed a SI LAI/FPAR CDR based on Terra-MODIS/Aqua-MODIS/VIIRS LAI/FPAR standard products
and ST-Tensor completion model. The CDR covers a substantial temporal period from 2000 to 2022, with spatial resolutions
of 500m/5km/0.05 degrees and temporal resolutions of 8 days or half month. The generation of the SI LAI/FPAR CDR was
evaluated at each step, including the evaluation of every step and ST-Tensor completion model. Evaluation results show that
the elimination of low-quality LAI/FPAR and consolidating Terra/Aqua/VIIRS LAI/FPAR into SI LAI/FPAR are effective in
the production process; and the ST-Tensor completion model was excelled in gap filling. The ground-based validations show
that the newly generated SI LAI/FPAR CDR achieves higher accuracy compared to original Terra/Aqua/VIIRS LAI/FPAR
products. Specifically, the SI LAI/FPAR CDR shows the highest accuracy ($R^2$ = 0.72/0.79 and RMSE = 0.84/0.15 for
LAI/FAPR) among all LAI/FPAR products with GBOV LAI/FPAR as benchmark. Similarly, the SI LAI/FPAR CDR shows
an increase $R^2$ magnitude of 0.04~0.16/0.05~0.09 and a decrease RMSE magnitude of 0.02~0.19/0.01~0.02 based on DIRECT

LAI/FPAR. The evaluation results also shows that the SI LAI/FPAR CDR has a lower TSS compared to the original Terra/Aqua/VIIRS LAI/FPAR product, which suggests that this CDR has less noise and provides more stable timeseries. Conversely, the MAE is also lower, indicating that the SI LAI/FPAR CDR is closer to high-quality LAI/FPAR retrievals compared to Terra/Aqua/VIIRS LAI/FPAR products. Additionally, the results of the 'Greening Earth' study demonstrate the consistency of trend analysis based on the SI LAI/FPAR CDR with previous findings. The same consistency analysis greatly enhances the ability to conduct further studies on vegetation dynamics and climate change. By exploiting the integration of multiple satellite data sources and applying advanced gap-filling techniques, the SI LAI/FPAR CDR presented in this study provides a valuable resource for researchers studying vegetation dynamics and their relationship to climate change. Overall, the rigorous evaluation and validation conducted throughout the study provides confidence in the accuracy and reliability of the SI LAI/FFAPR CDR, further strengthening its utility for diverse applications in environmental science and land management.

**Data availability.**

The SI LAI/FPAR CDR is openly available at: https://doi.org/10.5281/zenodo.8076540 (Pu et al., 2023a), https://code.earthengine.google.com/?asset=projects/sat-io/open-datasets/BU_LAI_FPAR/wgs_500m_8d, https://code.earthengine.google.com/?asset=projects/sat-io/open-datasets/BU_LAI_FPAR/wgs_500m_bimonthly.

And the Readme files about data description, data availability, and example code of GEE and MATLAB for SI LAI/FPAR CDR can be found at https://github.com/JiabinPu/Sensor-Independent-LAI-FPAR-CDR.

**CRediT Author Statement.**

**JP:** Methodology, Conceptualization, Software, Formal analysis, Writing - Original Draft. **KY:** Methodology, Conceptualization, Draft Revision, Supervision. **SR:** Dataset Management, Resources. **ZZ:** Draft Review and Revision. **MR:** Draft Review and Revision. **YK:** Formal analysis, Draft Revision, Supervision, Resources. **RM:** Conceptualization, Resources, Supervision.

**Competing Interests.**

The authors declare that they have no known competing financial interests or personal relationships that could have influenced the work reported in this study.



**Disclaimer.**

Publisher's note: Copernicus Publications remains neutral with regard to jurisdictional claims in published maps and institutional affiliations.

510 **Acknowledgments.**

We would like to thank NASA for providing MODIS&VIIRS standard products. We gratefully acknowledge the Google Earth Engine (https://earthengine.google.com/). We also like to thank Dong Chu in Wuhan University for the code of ST-Tensor completion model code.

**Financial Support.**

515 This work was supported by the NASA Grants to Boston University under the MODIS and VIIRS Projects (80NSSC21K1925 and 80NSSC21K1960).

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
