# Peer review of "Sensor-Independent LAI/FPAR CDR: Reconstructing a Global Sensor-Independent Climate Data Record of MODIS and VIIRS LAI/FPAR from 2000 to 2022"

_Earth System Science Data, 2023_

## Author Response (AR1)

(The reviewer's comments are in blue font while our responses use black font)

**Response to Reviewer 1**

Response: We would like to thank the reviewers for their careful and constructive reviews. These comments have played a pivotal role in enhancing the overall quality of this work. We have carefully revised our manuscript and adequately addressed all the questions and concerns that the referees have raised, and you can find our detailed responses in the attached document.

*The study proposes to generate a sensor-independent LAI and FPAR climate data record (CDR) based on existing global MODIS and VIIR data sets. A spatial-temporal tensor completion model was used to fill the gaps from the selected highest quality data. The paper is generally organized. The new data set would be useful for the Earth system science studies.*

Response: We thank the referee for his/her thoughts and comments. Our responses are given below.

*GENERAL COMMENTS*

*I have two major concerns.*

*1. I feel a bit lost with the huge difference (2~3) between the original and filtered LAIs in Fig. 3. I understand the filtered LAI is from the highest quality original data. It's difficult to understand the huge increase after filtering. Is this because of the small fraction of main algorithm used? Authors may show the percentage of main algorithms or the retrieval index in the time series.*

Response: Thanks for your suggestion. Yes, this is because of the small fraction of main algorithm retrievals. To better explain this, we have incorporated the Retrieval Index (RI) time series into these figures, as reviewer suggested. Now one can clearly see a negative correlation between the RI and the contrast in LAI/FPAR values before and after filtering. is most pronounced when the RI is low. Theoretically, this behavior can be explained by significant differences in the quality of retrievals produced by the main and backup algorithms. The following sentence has been added to the caption of Figure 3: "RI is negatively correlated with the difference between the original and filtered values, suggesting that this procedure reduces the impact of poor-quality retrievals on the time series."

[Figure]

**Figure 3.** The temporal comparisons between the original Terra/Aqua/VIIRS LAI and Filtered Terra/Aqua/VIIRS LAI for the EBF of Amazon Forest region. The panel (a)~(c) represent Terra/Aqua/VIIRS, respectively. The blue and purple lines mean original and filtered LAI (left y-axis). And the shadow indicates the RI (right y-axis). RI is negatively correlated with the difference between the original and filtered values, suggesting that this procedure reduces the impact of poor-quality retrievals on the time series.

[Figure]

**Figure S3.** Same as Fig. 3 but for FPAR. The temporal comparisons between the original Terra/Aqua/VIIRS FPAR and Filtered Terra/Aqua/VIIRS FPAR for the EBF of Amazon Forest region.

*2. I don't think Fig. 6 is useful to explain Step 3. Authors used 5% of the pixels to test the ST-tensor; this is an independent exercise that is not very related to the context of this study. As a matter of fact, the effect of the TS-tensor has been demonstrated in many other studies and was already adopted in this study.*

*Readers would be more interested to see: 1) the real % of missing pixels that were gap-filled by the TS-tensor; 2) whether the recovered pixels are spatial-temporally reasonable. Therefore, Fig. 6 can be exchanged with spatial-temporal maps before and after gap filling.*

Response: Note that Table 3 shows the real % of missing pixels that were gap-filled by the TS-tensor for the all sensors (column "No data") and sensor specific (100%-sensor specific %). As reviewer

suggested, Figure 6 has been replaced with a new one showing spatial-temporal LAI maps before and after gap filling. Same for Figure S4.

[Figure]

**Figure 6.** The spatial performance of the ST-Tensor method for DOY 1 (a), 89 (b), 177 (c), 281 (d) in 2014 year in the Amazon Forest region. Left panels and right panels represent the filtered SI LAI and ST-Tensor ST LAI, respectively.

[Figure]

**Figure S4.** Same as Fig. 6 but for FPAR. The spatial performance of the ST-Tensor method for DOY 1 (a), 89 (b), 177 (c), 281 (d) in 2014 year in the Amazon Forest region.

*SPECIFIC COMMENTS*

*L64-65. This sentence is confusing. Rephrase.*

Response: Thanks for your comment. This sentence has been revised to read "These methods would increase product spatial coverage. And the scientific community also developed various gap filling techniques to extrapolate retrievals beyond areas with valid satellite observations such as…"

*L136. Sentence incomplete.*

Response: Thanks for your comment. This sentence has been revised to read: "The only notable exceptions are the eastern part of China and Eastern Europe, which lacks the measurement sites."

*Eq. (4) The function sgn() was not explained.*

Response: Thanks for your suggestion. Its definition has been added as "The function sgn(x) takes the value 1 if its argument is a positive number, and -1 otherwise."

*L227. This filtered timeseries that are used as reference are crucial intermediate results. This reference data should be presented either in time series or with maps. This would greatly help readers understand Figs. 7 & 8 in section 4.2.*

Response: Thanks for this suggestion. We added the reference maps of LAI/FPAR for different DOYs (1, 89, 177, 281) to supplementary (Fig S1 and Fig S2, new figures in supplementary). It is notable that the SI LAI/FPAR CDR have the similar performance with the reference LAI/FPAR.

[Figure]

**Figure S1.** The spatial performance of the LAI reference for DOY 1 (a), 89 (b), 177 (c), 281 (d) in the Amazon Forest region.

[Figure]

**Figure S2.** Same as Fig. S1 but for FPAR. The spatial performance of the FPAR reference for DOY 1 (a), 89 (b), 177 (c), 281 (d) in the Amazon Forest region.

*L306-314. This texts are difficult to read because of the missing link between text and Table 3 (difficult to read). Suggest to rephrase the text and add a column showing the percentage of all high quality pixels that were presented in the texts.*

Response: Thank you for pointing this out. We have revised it as follows.

"Fig. 4 illustrates percentages of high-quality retrievals from single sensors and after step 2 for selected Amazon Forest region (zoom-in case in Fig. 1) for the 2013 to 2022 overlap period. Table 3 summarizes the changes after the implementation of step 2 for the whole and overlap periods. For the overlap period the percentages increase from 41.09% (Terra), 29.55% (Aqua), and 30.90% (VIIRS) to 58.50% in Filtered SI timeseries. If the time interval is extended to the whole dataset record period (2000 to 2022), the percentages increase from 41.68%, 26.73%, and 14.85% to 54.07%. Consequently, the percentage of pixels that need to be gap filled reaches 45.93%."

**Table 3.** Percentage of pixels with high-quality retrievals from single sensors and their combinations for whole (2000 to 2022) and overlap (2013 to 2022) periods.

| 2000 ~ 2022 |
| --- |
| 45.93% |
| 20.35% |
| 6.91% |
| 3.62% |
| 11.95% |
| 3.36% |
| 1.85% |
| 6.02% |
| 41.68% |
| 26.73% |
| 14.85% |
| 54.07% |

| | 2013 ~ 2022 |
| --- | --- |
| No data | 41.50% |
| Terra Only | 14.37% |
| Aqua Only | 5.94% |
| VIIRS Only | 7.60% |
| Terra & Aqua | 7.29% |
| Terra & VIIRS | 6.98% |
| Aqua & VIIRS | 3.87% |
| All Sensors | 12.45% |
| Terra Involved | 41.09% |
| Aqua Involved | 29.55% |
| VIRRS Involved | 30.90% |
| Filtered SI | 58.50% |

*Figs. 7 & 8 As noted above (L227), the reference maps may be displayed somewhere earlier.*

Response: Same as above.

*L494. (Pu et al., 2023a) is not in references.*

Response: Thanks, and we forgot to attach the link before. The "Pu et al., 2023a" should be "Pu, J., Roy, S., Knyazikhin, Y., & Myneni, R. B. (2023a). Sensor-Independent LAI/FPAR CDR [dataset]. Zenodo. https://doi.org/10.5281/zenodo.8076540". This is the public dataset that we uploaded to the Zenodo.

**Response to Reviewer 2**

*I have reviewed the paper, and I must acknowledge that it is well-written and comprehensive in addressing the development of a Sensor-Independent (SI) Leaf Area Index (LAI) and Fraction of Photosynthetically Active Radiation (FPAR) Climate Data Record (CDR). The study addresses the limitations of existing LAI/FPAR products and takes a significant step forward in improving their accuracy and reliability. Below are my key observations:*

*Strengths:*

*Clarity of Presentation: The paper is written in a clear and organized manner. It effectively communicates the significance of LAI and FPAR, the challenges faced, and the proposed solution.*
*Research Significance: The paper makes a strong case for the importance of LAI and FPAR in ecological and climate studies, providing a comprehensive background on their relevance.*
*Awareness of Weaknesses: The authors appropriately identify the weaknesses in existing LAI/FPAR products, emphasizing the need for a new approach.*
*Comprehensive Approach: The study outlines a detailed methodology, including sensor-specific quality assessment, ST-Tensor completion model, and various spatial and temporal resolutions, providing a holistic approach to CDR development.*
*Use of Citations: Proper citations of prior research strengthen the credibility of the paper and demonstrate a thorough literature review.*
*Areas that may be considered for further improvement:*

*Sentence Structure: A few complex sentences could be simplified to enhance readability, particularly for a broader audience.*
*Conclusion:*

*In summary, this paper is a commendable contribution to the field of LAI and FPAR data records. Its clear presentation, strong research significance, and comprehensive approach make it valuable. Addressing the minor points of improvement would enhance its accessibility and overall readability. I recommend it for publication with these suggestions in mind.*

> Response: On behalf of our co-authors, we would like to thank you for your constructive comments and suggestions, which we have addressed after careful consideration. We also extend our apologies for any editorial errors that have appeared in the manuscript. We have conducted thorough proofreading to enhance the manuscript's readability. Your recognition of our work, along with your support and encouragement, is greatly appreciated. Thank you once again.